# Bayesian Network Model Averaging Classifiers by Subbagging

**DOI:** 10.3390/e24050743

**Published:** 2022-05-23

**Authors:** Shouta Sugahara, Itsuki Aomi, Maomi Ueno

**Affiliations:** 1Graduate School of Informatics and Engineering, The University of Electro-Communications, 1-5-1, Chofugaoka, Chofu-shi 182-8585, Japan; ueno@ai.lab.uec.ac.jp; 2Sansan Inc., Tokyo 150-0001, Japan; aomi@sansan.com

**Keywords:** Bayesian networks, classification, model averaging, structure learning

## Abstract

When applied to classification problems, Bayesian networks are often used to infer a class variable when given feature variables. Earlier reports have described that the classification accuracy of Bayesian network structures achieved by maximizing the marginal likelihood (ML) is lower than that achieved by maximizing the conditional log likelihood (CLL) of a class variable given the feature variables. Nevertheless, because ML has asymptotic consistency, the performance of Bayesian network structures achieved by maximizing ML is not necessarily worse than that achieved by maximizing CLL for large data. However, the error of learning structures by maximizing the ML becomes much larger for small sample sizes. That large error degrades the classification accuracy. As a method to resolve this shortcoming, model averaging has been proposed to marginalize the class variable posterior over all structures. However, the posterior standard error of each structure in the model averaging becomes large as the sample size becomes small; it subsequently degrades the classification accuracy. The main idea of this study is to improve the classification accuracy using subbagging, which is modified bagging using random sampling without replacement, to reduce the posterior standard error of each structure in model averaging. Moreover, to guarantee asymptotic consistency, we use the *K*-best method with the ML score. The experimentally obtained results demonstrate that our proposed method provides more accurate classification than earlier BNC methods and the other state-of-the-art ensemble methods do.

## 1. Introduction

A Bayesian network is a graphical model that represents probabilistic relations among random variables as a directed acyclic graph (DAG). The Bayesian network provides a good approximation of a joint probability distribution because it decomposes the distribution exactly into a product of the conditional probabilities for each variable when the probability distribution has a DAG structure, depicted in Figure 1.

Bayesian network structures are generally unknown. For that reason, they must be estimated from observed data. This estimation problem is called Learning Bayesian networks. The most common learning approach is a score-based approach, which seeks the best structure that maximizes a score function. The most widely used score is the marginal likelihood for finding a maximum a posteriori structure [1,2]. The marginal likelihood (ML) score based on the Dirichlet prior ensuring likelihood equivalence is called Bayesian Dirichlet equivalence [2]. Given no prior knowledge, the Bayesian Dirichlet equivalence uniform (BDeu), as proposed by Buntine [1], is often used. These scores require an equivalent sample size (ESS), which is the value of a user-determined free parameter. As demonstrated in recent studies, ESS plays an important role in resulting network structure estimated using BDeu [3,4,5,6]. However, this approach has an associated NP-hard problem [7]: the number of structure searches increases exponentially with the number of variables.

Bayesian network classifiers (BNC), which are special cases of Bayesian networks designed for classification problems, have yielded good results in real-world applications, such as facial recognition [8] and medical data analysis [9]. The most common score for BNC structures is conditional log likelihood (CLL) of the class variable given all the feature variables [10,11,12]. Friedman et al. [10] claimed that the structure maximizing CLL, called a discriminative model, provides a more accurate classification than that maximizing the ML. The reason is that the CLL reflects only the class variable posterior, whereas the ML reflects the posteriors of all the variables.

Nevertheless, ML is known to have asymptotic consistency, which guarantees that the structure which maximizes the ML converges to the true structure when the sample size is sufficiently large. Therefore, Sugahara et al. [13], Sugahara and Ueno [14] demonstrated experimentally that the BNC performance achieved by maximizing the ML is not necessarily worse than that achieved by maximizing CLL for large data. Unfortunately, their experiments also demonstrated that the classification accuracy of the structure maximizing the ML rapidly worsens as the sample size becomes small. They explained the reason: the class variable tends to have numerous parents when the sample size is small. Therefore, the conditional probability parameter estimation of the class variable becomes unstable because the number of parent configurations becomes large. Then the sample size for learning a parameter becomes sparse. This analysis suggests that exact learning BNC by maximizing the ML to have no parents of the class variable might improve the classification accuracy. Consequently, they proposed exact learning augmented naive Bayes classifier (ANB), in which the class variable has no parent and in which all feature variables have the class variable as a parent. Additionally, they demonstrated the effectiveness of their method empirically.

The salient reason for difficulties with this method is that the error of learning structures becomes large when the sample size becomes small. Model averaging, which marginalizes the class variable posterior over all structures, has been regarded as a method to alleviate this shortcoming [15,16,17,18]. However, the model averaging approach confronts the difficulty that the number of structures increases super-exponentially for the network size. Therefore, averaging all structures with numerous variables is computationally infeasible. The most common approach to this difficulty is the *K*-best method [19,20,21,22,23,24,25], which considers only the *K*-best scoring structures.

Another point of difficulty is that the posterior standard error of each structure in the model averaging becomes large for a small sample size; it then decreases the classification accuracy. As methods to reduce the posterior standard error, resampling methods such as the adaboost (adaptive boosting) [26], bagging (bootstrap aggregating) [27], and subbagging (subsampling aggregating) [28] are known. In addition, Jing et al. [29] proposed ensemble class variable prediction using adaboost. That study demonstrated its effectiveness empirically. However, this method tends to cause overfitting for small data because adaboost tends to be sensitive to noisy data [30]. Later, Rohekar et al. [31] proposed B-RAI, a model averaging method with bagging, based on the RAI algorithm [32], which learns a structure by recursively conducting conditional independence (CI) tests, edge direction, and structure decomposition into smaller substructures. The B-RAI increases the number of models for model averaging using multiple bootstrapped datasets. However, B-RAI is inapplicable for bagging to the posterior of the structures. Therefore, the posterior standard error of the structures is not expected to decrease. In addition, the CI tests of the B-RAI are not guaranteed to have asymptotic consistency. Although B-RAI reduces the computational costs, it degrades the classification accuracy for large data.

The main idea of this study is to improve the classification accuracy using the subbagging to reduce the posterior standard error of structures in model averaging. Moreover, to guarantee asymptotic consistency, we use the *K*-best method with the ML score. The proposed method, which we call Subbagging *K*-best method (SubbKB), is expected to present the following benefits: (1) Because the SubbKB has an asymptotic consistency for the true class variable classification, the class variable posterior converges to the true value when the sample size becomes sufficiently large; (2) even for small data, subbagging reduces the posterior standard error of each structure in the *K*-best structures and improves the classification accuracy. For this study, we conducted experiments to compare the respective classification performances of our method and earlier methods. Results of those experiments demonstrate that SubbKB provides more accurate classification than the earlier BNC methods do. Furthermore, our experiments compare SubbKB and the other state-of-the-art ensemble methods. Results indicate that SubbKB outperforms the state-of-the-art ensemble methods.

This paper is organized as follows. In Section 2 we review Bayesian networks and BNCs. In Section 3 we review model averaging of BNCs. In Section 4 we describe SubbKB and prove that SubbKB asymptotically estimates a true conditional probability. In Section 5 we describe in detail the experimental setup and results. Finally, we conclude in Section 6.

## 2. Bayesian Network Classifier

### 2.1. Bayesian Network

Letting $\left\{X_{0},X_{1},\cdots ,X_{n}\right\}$ be a set of n+1 discrete variables, then $X_{i},\left(i=0,\cdots ,n\right)$ can take values in the set of states $\left\{1,\cdots ,r_{i}\right\}$. One can write $X_{i}=k$ when observing that an $X_{i}$ is state *k*. According to the Bayesian network structure *G*, the joint probability distribution is $P\left(X_{0},X_{1},\cdots ,X_{n}\right)=\prod_{i=0}^{n}P\left(X_{i}|\Pi_{i},G\right),$ where $\Pi_{i}$ is the parent variable set of $X_{i}$. Letting $\theta_{ijk}$ be a conditional probability parameter of $X_{i}=k$ when the *j*-th instance of the parents of $X_{i}$ is observed (we write $\Pi_{i}=j$), we define $\Theta =\left\{\theta_{ijk}\right\}\;\left(i=0,\cdots ,n;j=1,\cdots ,q_{i};k=1,\cdots ,r_{i}\right)$. A Bayesian network is a pair B=(G,Θ).

The Bayesian network (BN) structure represents conditional independence assertions in the probability distribution by *d-separation*. First, we define *collider*, for which we need to define the d-separation. Letting *path* denote a sequence of adjacent variables, the collider is defined as shown below.

**Definition** **1.**
*Assuming a structure G=(V,E), a variable Z∈V on a path ρ is a collider if and only if there exist two distinct incoming edges into Z from non-adjacent variables.*


We then define d-separation as explained below.

**Definition** **2.**
*Assuming we have a structure G=(V,E), X,Y∈V, and Z⊆V∖{X,Y}, the two variables X and Y are d-separated, given Z in G, if and only if every path ρ between X and Y satisfies either of the following two conditions:*

*Z includes a non-collider on ρ.*

*There is a collider Z on ρ, and Z does not include Z or its descendants.*

*We denote the d-separation between X and Y given Z in the structure G as $Dsep_{G}\left(X,Y|Z\right)$. Two variables are d-connected if they are not d-separated.*


Let I(X,Y∣Z) denote that *X* and *Y* are conditionally independent given Z in the true joint probability distribution. A BN structure *G* is an *independence map (I-map)* if all d-separations in *G* are entailed by conditional independence in the true probability distribution.

**Definition** **3.**
*Assuming a structure G=(V,E), X,Y∈V, and Z⊆V∖{X,Y}, then G is an I-map if the following proposition holds:*

$$\begin{array}{c} \forall X,Y\in V,\forall Z\subseteq V\setminus \left\{X,Y\right\},Dsep_{G}\left(X,Y|Z\right)\Rightarrow I\left(X,Y|Z\right). \end{array}$$



Probability distributions represented by an I-map converge to the true probability distribution when the sample size becomes sufficiently large.

For an earlier study, Buntine [1] assumed the Dirichlet prior and used an expected a posteriori (EAP) estimator ${\hat{\theta }}_{ijk}=\left(\alpha_{ijk}+N_{ijk}\right)/\left(\alpha_{ij}+N_{ij}\right).$ In that equation, $N_{ijk}$ represents the number of samples of $X_{i}=k$ when $\Pi_{i}=j$, $N_{ij}=\sum_{k=1}^{r_{i}}N_{ijk}$. Additionally, $\alpha_{ijk}$ denotes the hyperparameters of the Dirichlet prior distributions ($\alpha_{ijk}$ as a pseudo-sample corresponding to $N_{ijk}$); $\alpha_{ij}=\sum_{k=1}^{r_{i}}\alpha_{ijk}$.

The first learning task of the Bayesian network is to seek a structure *G* that optimizes a given score. Let $D=\{u^{1},\cdots ,u^{d},\cdots ,u^{N}\}$ be training dataset. In addition, let each $u^{d}$ be a tuple of the form $\langle x_{0}^{d},x_{1}^{d},\cdots ,x_{n}^{d}\rangle$. The most popular marginal likelihood (ML) score, P(D∣G), of the Bayesian network finds the maximum a posteriori (MAP) structure $G^{*}$ when we assume a uniform prior P(G) over structures, as presented below.
$$\begin{array}{c} G^{*}=\underset{G}{\arg\;\max}P\left(G|D\right)=\underset{G}{\arg\;\max}\frac{P(D|G)P(G)}{P(D)}=\underset{G}{\arg\;\max}P\left(D|G\right). \end{array}$$
The ML has an asymptotic consistency [33], i.e., the structure which maximizes ML converges to the true structure when the sample is large. In addition, the Dirichlet prior is known as a distribution that ensures likelihood equivalence. This score is known as Bayesian Dirichlet equivalence [2]. Given no prior knowledge, the Bayesian Dirichlet equivalence uniform (BDeu), as proposed earlier by Buntine [1], is often used. The BDeu score is represented as:$$\begin{array}{cc} P\left(D|G\right)=\prod_{i=0}^{n}\prod_{j=1}^{q_{i}} & \frac{\Gamma (\frac{\alpha }{q_{i}})}{\Gamma (\frac{\alpha }{q_{i}}+N_{ij})}\prod_{k=1}^{r_{i}}\frac{\Gamma (\frac{\alpha }{r_{i}q_{i}}+N_{ijk})}{\Gamma (\frac{\alpha }{r_{i}q_{i}})}, \end{array}$$
where α denotes a hyperparameter. Earlier studies [5,6,34,35] have demonstrated that learning structures are highly sensitive to α. Those studies demonstrated α=1.0 as the best method to mitigate the influence of α for parameter estimation.

### 2.2. Bayesian Network Classifiers

A Bayesian network classifier (BNC) can be interpreted as a Bayesian network for which $X_{0}$ is the class variable and for which $X_{1},\cdots ,X_{n}$ are feature variables. Given an instance $x=\langle x_{1},\cdots ,x_{n}\rangle$ for feature variables $X_{1},\ldots ,X_{n}$, BNC *B* predicts the class variable’s value by maximizing the posterior as $\hat{c}={\mathrm{argmax}}_{c\in \{1,\cdots ,r_{0}\}}P\left(c|x,B\right).$

However, Friedman et al. [10] reported the BNC maximizing ML as unable to optimize the classification performance. They proposed the sole use of the conditional log likelihood (CLL) of the class variable given the feature variables instead of the log likelihood for learning BNC structures.

Unfortunately, no closed-form formula exists for optimal parameter estimates to maximize CLL. Therefore, for each structure candidate, learning the network structure maximizing CLL requires the use of some search methods such as gradient descent over the space of parameters. For that reason, the exact learning of network structures by maximizing CLL is computationally infeasible.

As a simple means of resolving this difficulty, Friedman et al. [10] proposed the tree-augmented naive Bayes (TAN) classifier, for which the class variable has no parent and for which each feature variable has a class variable and at most one other feature variable as a parent variable.

In addition, Carvalho et al. [12], Carvalho et al. [36] proposed an approximated conditional log likelihood (aCLL) score, which is both decomposable and computationally efficient. Letting $G_{ANB}$ be an ANB structure, we define $\Pi_{i}^{*}=\Pi_{i}\setminus \left\{X_{0}\right\}$ based on $G_{ANB}$. Additionally, we let $N_{ijck}$ be the number of samples of $X_{i}=k$ when $X_{0}=c$ and $\Pi_{i}^{*}=j$$\left(i=1,\cdots ,n;j=1,\cdots ,q_{i}^{*};c=1,\cdots ,r_{0};k=1,\cdots ,r_{i}\right)$. Moreover, we let $N^{\prime }>0$ represent the number of pseudo-counts. Under several assumptions, aCLL can be represented as:$$\begin{array}{c} aCLL\left(G_{ANB}|D\right)\propto \sum_{i=1}^{n}\sum_{j=1}^{q_{i}^{*}}\sum_{k=1}^{r_{i}}\sum_{c=1}^{r_{0}}\left(N_{ijck}+\beta \sum_{c^{\prime }=1}^{r_{0}}N_{ijc^{\prime }k}\right)\log\frac{N_{ij+ck}}{N_{ij+c}}, \end{array}$$
where
$$N_{ij+ck}=\left\{\begin{array}{c} N_{ijck}+\beta \sum_{c^{\prime }=1}^{r_{0}}N_{ijc^{\prime }k}\;\;\;\mathrm{if}\;N_{ijck}+\beta \sum_{c^{\prime }=1}^{r_{0}}N_{ijc^{\prime }k}\geq N^{\prime }\\ N^{\prime }\;\;\;\;\;\;\;\;\;\;\;\;\;\;\;\;\;\;\;\;\;\;\;\;\;\;\;\;\mathrm{otherwise}, \end{array}\right.$$
$$N_{ij+c}=\sum_{k=1}^{r_{i}}N_{ij+ck}.$$
The value of β is found using Monte Carlo method to approximate CLL. When the value of β is optimal, then aCLL is a minimum-variance unbiased approximation of CLL. A report of an earlier study described that the classifier maximizing the approximated CLL provides a better performance than that maximizing the approximated ML.

Unfortunately, they stated no reason for why CLL outperformed ML. Differences of performance between ML and CLL in earlier studies might depend on the learning algorithms which were employed because they used not exact but approximate learning algorithms. Therefore, Sugahara et al. [13], Sugahara and Ueno [14] demonstrated by experimentation that the BNC performance achieved by maximizing the ML is not necessarily worse than that achieved by maximizing CLL for large data, although both performances may depend on the nature of the dataset. However, the classification accuracy of the structure maximizing the ML becomes rapidly worse as the sample size becomes small.

## 3. Model Averaging of Bayesian Network Classifiers

The less accurate classification of BNCs for small data results from learning structure errors. As a method to alleviate this shortcoming, model averaging, which marginalizes the class variable posterior over all structures, is reportedly effective [15,16,17,18]. Using model averaging, the class variable’s value *c* is estimated as:$$\begin{array}{cc} \hat{c} & =\underset{c\in \{1,\cdots ,r_{0}\}}{\arg\;\max}P\left(c|x,D\right)\\ \; & =\underset{c\in \{1,\cdots ,r_{0}\}}{\arg\;\max}\sum_{G\in G}P\left(G|D\right)P\left(c|x,G,D\right)\\ \; & =\underset{c\in \{1,\cdots ,r_{0}\}}{\arg\;\max}\sum_{G\in G}P\left(D|G\right)P\left(c|x,G,D\right), \end{array}$$
where G is a set of all structures. However, the number of structures increases super-exponentially for the network size. Therefore, averaging all the structures with numerous variables is computationally infeasible. The most commonly used approach to resolve this problem is a *K*-best structures method [19,21,22,23,24,25], which considers only the *K*-best scoring structures. However, the *K*-best structures method finds the best *K* individual structures included in Markov equivalence classes, where the structures within each class represent the same set of conditional independence assertions and determine the same statistical model. To address the difficulty, Chen and Tian [20] proposed the *K*-best EC method, which can be used to ascertain the *K* best equivalence classes directly. These methods have asymptotic consistency if they use an exact learning algorithm. Using the *K*-best scoring structures, ${\left\{G^{k}\right\}}_{k=1}^{K}$, the class variable posterior can be approximated as:$$P\left(X_{0}|x,D\right)\approx \sum_{k=1}^{K}\frac{P(D|G^{k})}{\sum_{k^{\prime }=1}^{K}P\left(D|G^{k^{\prime }}\right)}P\left(X_{0}|x,G^{k},D\right).$$

The posterior standard error of each structure in the model averaging becomes large as the sample size becomes small; as it does so, it decreases the classification accuracy. However, resampling methods such as adaboost [26] and bagging [27] are known to reduce the standard error of estimation. Actually, Jing et al. [29] proposed the $bAN_{mix}$ boosting method, which predicts the class variable using adaboost. Nevertheless, this method is not a model averaging method. It tends to cause overfitting for small data because adaboost tends to be sensitive to noisy data [30].

Rohekar et al. [31] proposed a model averaging method named B-RAI, based on the RAI algorithm [32]. The method learns the structure by sequential application of conditional independence (CI) tests, edge direction and structure decomposition into smaller substructures. This sequence of operations is performed recursively for each substructure, along with increasing order of the CI tests. At each level of recursion, the current structure is first refined by removing edges between variables that are independently conditioned on a set of size $n_{z}$, with subsequent directing of the edges. Then, the structure is decomposed into ancestors and descendant groups. Each group is autonomous: it includes the parents of its members [32]. Furthermore, each autonomous group from the $n_{z}$-th recursion level is partitioned independently, resulting in a new level of $n_{z}+1$. Each such structure (a substructure over the autonomous set) is partitioned progressively (in a recursive manner) until a termination condition is satisfied (independence tests with condition set size $n_{z}$ cannot be performed), at which point the resulting structure (a substructure) at that level is returned to its parent (the previous recursive call). Similarly, each group in its turn, at each recursion level, gathers back the structures (substructures) from the recursion level that followed it; it then returns itself to the recursion level that precedes it until the highest recursion level $n_{z}=0$ is reached and the final structure is fully constructed. Consequently, RAI constructs a tree in which each node represents a substructure and for which the level of the node corresponds to the maximal order of conditional independence that is encoded in the structure. Based on the RAI algorithm, B-RAI constructs a structure tree from which structures can be sampled. In essence, it replaces each node in the execution tree of the RAI with a bootstrap node. In the bootstrap node, for each autonomous group, *s* datasets are sampled with replacement from training data *D*. They calculate log[P(D∣G)] for each leaf node in the tree (*G* is the structure in the leaf) using the BDeu score. For each autonomous group, given *s* sampled structures and their scores returned from *s* recursive calls, the B-RAI samples one of the *s* results proportionally to their (log) score. Finally, the sampled structures are merged. The sum of scores of all autonomous sets is the score of the merged structure.

However, B-RAI does not apply bagging to the posterior of the structures. Therefore, the posterior standard error of each structure is not expected to decrease. In addition, the B-RAI is not guaranteed to have asymptotic consistency. This lack of consistency engenders the reduction of the computational costs, but degradation of the classification accuracy.

## 4. Proposed Method

This section presents the proposed method, SubbKB, which improves the classification accuracy using resampling methods to reduce the posterior standard error of each structure in model averaging. As described in Section 3, the existing resampling methods are not expected to reduce the posterior standard error of each structure in model averaging sufficiently. A simple method to resolve this difficulty might be a bagging using random sampling with replacement. However, sampling with replacement might increase the standard error of estimation as the sample size becomes small because of duplicated sampling [37]. To avoid this difficulty, we use subbagging [28], which is modified bagging using random sampling without replacement.

Consequently, SubbKB includes the following four steps: (1) Sample *T* datasets ${\left\{D_{t}\right\}}_{t=1}^{T}$ without replacement from training data *D*, where $|D_{t}|=\delta |D|,\left(0<\delta <1\right)$; (2) Learn the *K*-best BDeu scoring structures ${\left\{G_{t}^{k}\right\}}_{k=1}^{K}$, representing the *K*-best equivalence classes for each dataset $D_{t}$; (3) Compute the *T* class variable posteriors using each dataset $D_{t}$ and each set of *K*-best structures to model averaging; (4) By averaging the computed *T* class variable posteriors, the resulting conditional probability of the class variable given the feature variables is estimated as:(1)$$\begin{array}{c} P\left(X_{0}|x,D\right)\approx \frac{1}{T}\sum_{t=1}^{T}\sum_{k=1}^{K}\frac{P(D_{t}|G_{t}^{k})}{\sum_{k^{\prime }=1}^{K}P\left(D_{t}|G_{t}^{k^{\prime }}\right)}P\left(X_{0}|x,G_{t}^{k},D_{t}\right). \end{array}$$

The following theorem about SubbKB can be proved.

**Theorem** **1.**
*The SubbKB asymptotically estimates the true conditional probability of the class variable given the feature variables.*


**Proof.** From the asymptotic consistency of BDeu [38], the highest BDeu scoring structure converges asymptotically to the I-map with the fewest parameters denoted by $G^{*}$. Moreover, the ratio $P\left(D_{t}|G^{*}\right)/\sum_{k=1}^{K}P\left(D_{t}|G_{t}^{k}\right)$ asymptotically approaches 1.0. Therefore, Equation (Equation 1) converges asymptotically to
(2)$$\begin{array}{c} \frac{1}{T}\sum_{t=1}^{T}P\left(X_{0}|x,G^{*},D_{t}\right). \end{array}$$It is guaranteed that $P(X_{0}|x,G^{*},D_{t})$ converges asymptotically to the true conditional probability of the class variable given the feature variables denoted by $P^{*}\left(X_{0}|x\right)$. Consequently, formula (2) converges asymptotically to $P^{*}\left(X_{0}|x\right)$, i.e., SubbKB asymptotically estimates $P^{*}\left(X_{0}|x\right)$. □

The SubbKB is expected to provide the following benefits: (1) From Theorem 1, SubbKB asymptotically estimates the true conditional probability of the class variable given the feature variables; (2) For small data, subbagging reduces the posterior standard error of each structure learned using the *K*-best EC method and improves the classification accuracy. The next section explains experiments conducted to compare the respective classification performances of SubbKB and earlier methods.

## 5. Experiments

This section presents experiments comparing SubbKB and other existing methods.

### 5.1. Comparison of the SubbKB and Other Learning BNC Methods

First, we compare the classification accuracy of the following 14 methods:*NB*: Naive Bayes;*TAN* [10]: Tree-augmented naive Bayes;*aCLL-TAN* [12]: Exact learning TAN method by maximizing aCLL;*EBN*: Exact learning Bayesian network method by maximizing BDeu;*EANB*: Exact learning ANB method by maximizing BDeu;$bAN_{mix}$ [29]: Ensemble method using adaboost, which starts with naive Bayes and greedily augments the current structure at iteration *j* with the *j*-th edge having the highest conditional mutual information;*Adaboost(EBN)*: Ensemble method of 10 structures learned using adaboost to *EBN*;*B-RAI* [31]: Model averaging method over 100 structures sampled using B-RAI with s=3;*Bagging(EBN)*: Ensemble method of 10 structures learned using bagging to *EBN*;*Bagging(EANB)*: Ensemble method of 10 structures learned using bagging to *EANB*;*KB10* [20]: *K*-best EC method using the BDeu score with K=10;*KB10(EANB)*: *K*-best EC method under ANB constraints using the BDeu score with K=10;*KB20* [20]: *K*-best EC method using the BDeu score with K=20;*KB50* [20]: *K*-best EC method using the BDeu score with K=50;*KB100* [20]: *K*-best EC method using the BDeu score with K=100;*SubbKB10*: SubbKB with K=10 and T=10;*SubbKB10(MDL)*: the modified *SubbKB10* to use MDL score.
Here, the classification accuracy represents the average percentage correct among all classifications from ten-fold cross validation. Although determination of hyperparameter α of BDeu is difficult, we used α=1.0, which allows the data to reflect the estimated parameters to the greatest degree possible [5,6,34,35]. Note that α=1.0 is not guaranteed to provide optimal classification. We also used EAP estimators with $\alpha_{ijk}=1/\left(r_{i}q_{i}\right)$ as conditional probability parameters of the respective classifiers. Using *SubbKB10*, *Bagging(EBN)*, and *Bagging(EANB)*, the size of the sampled data is 90% of the training data. Our experiments were conducted entirely using a computational environment, as shown in Table 1. We used 26 classification benchmark datasets from the UCI repository, as shown in Table 2. Continuous variables were discretized into two bins using the median value as cut-off. Furthermore, data with missing values were removed from the datasets. Table 2 also presents the entropy of the class variable, $H(X_{0})$, for each dataset. For the discussion presented in this section, we define small datasets as datasets with less than 1000 sample size. In addition, we define large datasets as datasets with 1000 and more sample size. Throughout our experiments, we employ the ten-fold cross validation to evaluate methods.

Table 3 presents the classification accuracies of each method. The values shown in bold in Table 3 represent the best classification accuracies for each dataset. Moreover, we highlight the results obtained by the *SubbKB10* using a blue color. To confirm significance of the differences that arise when using *SubbKB10* and other methods, we applied the Hommel test [39], which is a non-parametric post hoc test used as a standard in machine learning studies [40]. Table 4 presents the *p*-values obtained using Hommel tests. From Table 3, results show that, among the methods explained above, the *SubbKB10* yields the best average accuracy. Moreover, from Table 4, *SubbKB10* outperforms all the model selection methods at the p<0.10 significance level. Particularly *NB*, *TAN*, and *aCLL-TAN* provide lower classification accuracy than the *SubbKB10* does for the No. 1, No. 20, and No. 24 datasets. The reason is that those methods have a small upper bound of maximum number of parents. Such a small upper bound is known to cause poor representational power of the structure [41]. The classification accuracy of *EBN* is the same as, or almost identical to, that of *SubbKB10* for large datasets such as No. 20, No. 23, No. 25, and No. 26 datasets because both methods have asymptotic consistency. However, the classification accuracy of *SubbKB10* is equal to or greater than that of *EBN* for small datasets from No. 1 to No. 15. As described previously, the classification accuracy of *EBN* is worse than that of the model averaging methods because the error of learning structure by *EBN* becomes large as the sample size becomes small.

For almost small datasets such as datasets from No. 6 to No. 9 and from No. 11 to No. 13, *SubbKB10* provides higher classification accuracy than *EANB* does because the error of learning ANB structures becomes large. However, for the No. 4 and No. 5 datasets, the classification accuracy of *EANB* is much higher than that obtained using *SubbKB10*. To analyze this phenomenon, we investigate the average number of the class variable’s parents in the structures learned by *EBN* and that by *SubbKB10*. The results displayed in Table 5 highlight that the average number of the class variable’s parents in the structures learned by *EBN* and that by *SubbKB10* tends to be large in the No. 4 and No. 5 datasets. Consequently, estimation of the conditional probability parameters of the class variable becomes unstable because the number of parent configurations becomes large. Then the sample size for learning a parameter becomes sparse. Actually, the ANB constraint prevents numerous parents of the class variable. Moreover, it improves the classification accuracy.

The *SubbKB10* outperforms $bAN_{mix}$, *Adaboost(EBN)*, *B-RAI*, *Bagging(EBN)*, *KB10*, *KB20*, and *KB50* at the *p* < 0.10 significance level. Actually, $bAN_{mix}$ provides much lower accuracy than *SubbKB10* does for the No. 1, No. 20, and No. 24 datasets because it has a small upper bound of a maximum number of parents, similar to *NB*, *TAN*, and *aCLL-TAN*. For almost all large datasets, the classification accuracy of *SubbKB10* is higher than that of *B-RAI* because *SubbKB10* has an asymptotic consistency, whereas *B-RAI* does not. The *SubbKB10* provides higher classification accuracy than *Adaboost(EBN)* does for small datasets, such as No. 5 and No. 10 datasets, because *Adaboost(EBN)* tends to cause overfitting, as described in Section 3. The classification accuracy of *Bagging(EBN)* is much worse than that of *SubbKB10* in the No. 5 dataset because the error of learning structures using each sampled data becomes large as the sample size becomes small. The *SubbKB10* alleviates this difficulty somewhat using model averaging for sampled data.

The *K*-best EC method using the BDeu score provides higher average classification accuracy as *K* increases, as shown by *EBN*, *KB10*, *KB20*, *KB50*, and *KB100*. Although the difference between *SubbKB10* and *KB100* is not statistically significant, *SubbKB10* provides higher average classification accuracy than *KB100* does. Moreover, we compare the classification accuracy of *SubbKB10* and the *K*-best EC methods using 1/100-sized subsamples from MAGICGT (No.26 of datasets) to confirm the robustness of *SubbKB10*. The results presented in Table 6 show that *SubbKB10* provides higher classification accuracy than the other *K*-best EC methods do.

Although *EANB* provides higher average classification accuracy than *EBN* does, *Bagging(EBN)* and *KB10* provides higher average accuracy than *Bagging(EANB)* and *KB10(EANB)* do, respectively. These results imply that the ANB constraint might not work effectively in model averaging because it decreases the diversity of the models; all the ANB structures necessarily have the edges between the class variable and all the feature variables. To confirm the diversity of ANB structures in model averaging, we compare the structural hamming distance (SHD) [42] of structures in each model averaging method. Table 7 presents the average SHDs between all two structures in each model averaging method. Results show that the average SHDs of *Bagging(EBN)* and *KB10* are higher than those of *Bagging(EANB)* and *KB10(EANB)*. That is, model averaging with ANB constraint has less diverse structures than that without ANB constraint does. Moreover, *SubbKB10* provides the highest average SHD among all the compared methods because the combination of *K*-best method and subbagging diversifies structures in the model averaging. *SubbKB10* provides higher average classification accuracy than *SubbKB10(MDL)* does. However, the difference between both methods is not statistically significant because the MDL score asymptotically converges to minus log BDeu score.

*SubbKB10* provides the highest accuracy among all the methods for the No. 22 dataset, which has the most entropy for the class variable among all the datasets. From Theorem 1, *SubbKB10* asymptotically estimates the true conditional probability of the class variable given the feature variables. Therefore, *SubbKB10* guarantees to provide high classification accuracy when the sample size is sufficiently large regardless of the distribution of the dataset.

Next, to demonstrate the advantages of using *SubbKB10* for small data, we compare the posterior standard error of the structures learned using *SubbKB10* with that learned by *KB100*. We estimate the posterior standard error of structures learned by the *KB100* as explained below.

Generate 10 random structures ${\left\{G_{m}\right\}}_{m=1}^{10}$;Sample 10 datasets, ${\left\{{\tilde{D}}_{i}\right\}}_{i=1}^{10}$, with replacement from the training dataset *D*, where $|{\tilde{D}}_{i}\left|=|D\right|$;Compute the posteriors $P\left(G_{m}|{\tilde{D}}_{i}\right)\approx P\left({\tilde{D}}_{i}|G_{m}\right)/\sum_{m^{\prime }=1}^{10}P\left({\tilde{D}}_{i}|G_{m^{\prime }}\right),\left(m=1,\cdots ,10;i=1,\cdots ,10\right)$;Estimate the standard error of the posteriors $P\left(G_{m}|D\right),\left(m=1,\cdots ,10\right)$ as:


(3)
$$\sqrt{\frac{1}{10(10-1)}\sum_{i=1}^{10}{\left\{P\left(G_{m}|{\tilde{D}}_{i}\right)-\frac{1}{10}\sum_{j=1}^{10}P\left(G_{m}|{\tilde{D}}_{j}\right)\right\}}^{2}}.$$


We estimate the posterior standard error of structures learned using *SubbKB10* as presented below.

Generate 10 random structures ${\left\{G_{m}\right\}}_{m=1}^{10}$;Sample 10 datasets, ${\left\{{\tilde{D}}_{ti}\right\}}_{i=1}^{10}$, with replacement from each bootstrapped dataset $D_{t}$, where $|{\tilde{D}}_{ti}\left|=\right|D_{t}|$;Compute the posteriors $P\left(G_{m}|{\tilde{D}}_{i}\right)\approx \frac{1}{T}\sum_{t=1}^{T}\left[P\left({\tilde{D}}_{it}|G_{m}\right)/\sum_{m^{\prime }=1}^{10}P\left({\tilde{D}}_{it}|G_{m^{\prime }}\right)\right],\left(m=1,\cdots ,10;i=1,\cdots ,10\right)$;Estimate the standard error of each of the posteriors $P\left(G_{m}|D\right),\left(m=1,\cdots ,10\right)$ using formula (3).

Average posterior standard errors over 10 structures ${\left\{G_{m}\right\}}_{m=1}^{10}$ of the *SubbKB10* and those of the *KB100* are presented in “APSES” of Table 8. Significance can be assessed from values obtained using the Wilcoxon signed-rank test. The *p*-values of the test are presented at the bottom of Table 8. Moreover, Table 8 presents the classification accuracies of the *KB100* and the *SubbKB10*. The results demonstrate that the APSES of *SubbKB10* is significantly lower than that of the *KB100*. Moreover, we investigate the relation between the APSES and the training data sample size. The APSES values of *SubbKB10* and *KB100* for the sample size are plotted in Figure 2. As presented in Figure 2, the APSESs of *KB100* are large for small sample size. As the sample size becomes large, the APSES of *KB100* becomes small and closes to that of *SubbKB10*. On the other hand, the APSESs of *SubbKB10* are constantly small independently of the sample size. As presented particularly in Table 8, *SubbKB10* provides higher classification accuracy than *KB100* does when the APSES of *SubbKB10* is lower than that of the *KB100*, such as that of No. 2, No. 6, and No. 9 datasets. Consequently, *SubbKB10* reduces the posterior standard error of the structures. It therefore improves the classification accuracy.

### 5.2. Comparison of *SubbKB10* and State-of-the-Art Ensemble Methods

This subsection presents a comparison of the classification accuracies of SubbKB10 with K=10,T=10 and state-of-the-art ensemble methods, i.e., XGBoost [23], CatBoost [43], and LightGBM [44]. Experimental setup and evaluation methods are the same as those of the previous subsection. We determine the ESS α∈{1,4,16,64,256,1024} in *SubbKB10* using ten-fold cross validation to obtain the highest classification accuracy. The six ESS values of α are determined according to Heckerman et al. [2]. To assess the significance of differences of *SubbKB10* from the other ensemble methods, we applied multiple Hommel tests [39]. Table 9 presents the classification accuracy and *p*-values obtained using Hommel tests. Results show that the differences between *SubbKB10* and any other ensemble methods are not statistically significant at the *p* < 0.10 level. However, *SubbKB10* provides higher average accuracy than XGBoost, CatBoost, and LightGBM do. CatBoost and LightGBM provides much worse accuracies than the other methods do for the No. 3 and No. 2 datasets, respectively. On the other hand, *SubbKB10* avoids to provide much worse accuracy than the other methods for small data because subbagging in *SubbKB10* improves the accuracy for small data, as previously demonstrated. Moreover, *SubbKB10* provides much higher accuracy than the other methods even for large data, the No. 24 dataset. The reason is that *SubbKB10* asymptotically estimates the true conditional probability of the class variable given the feature variables from Theorem 1. This property is also highly useful for developing AI systems based on decision theory because such systems require accurate probability estimations to calculate expected utility and loss [45,46].

## 6. Conclusions

This paper described our proposed subbagging of *K*-best EC to reduce the posterior standard error of each structure in model averaging. The class variable posterior of SubbKB converges to the true value when the sample size becomes sufficiently large because SubbKB has asymptotic consistency for true class variable classification. In addition, even for small data, SubbKB reduces the posterior standard error of each structure in the *K*-best structures and thereby improves the classification accuracy. Our experiments demonstrated that SubbKB provided more accurate classification than the *K*-best EC method and the other state-of-the-art ensemble methods did.

The SubbKB cannot learn large size of networks because of its large computational cost. We plan on exploring the following in future work:Steck and Jaakkola [47] proposed a conditional independence test with an asymptotic consistency, a Bayes factor with BDeu; Moreover, Abellán et al. [48], Natori et al. [49], Natori et al. [50] proposed constraint-based learning methods using a Bayes factor, which can learn large size of networks. We will apply the constraint-based learning methods using a Bayes factor to SubbKB so as to handle much larger number of variables in our method;Liao et al. [25] proposed a novel approach to model averaging Bayesian networks using a Bayes factor. Their approach is significantly more efficient and scales to much larger Bayesian networks than existing approaches. We expect to employ their method to address much larger number of variables in our method.Isozaki et al. [51], Isozaki et al. [52], Isozaki and Ueno [53] proposed an effective learning Bayesian network method by adjusting the hyperparameter for small data. We expect to employ their method instead of the BDeu to improve the classification accuracy for small data.

## Figures and Tables

**Figure 1 entropy-24-00743-f001:**
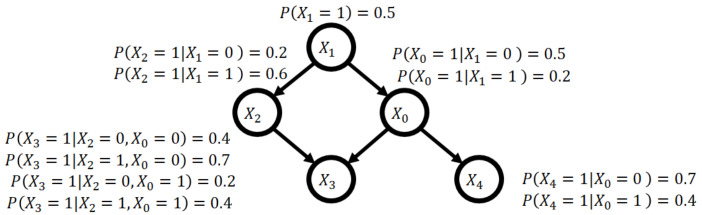
Example of a Bayesian network.

**Figure 2 entropy-24-00743-f002:**
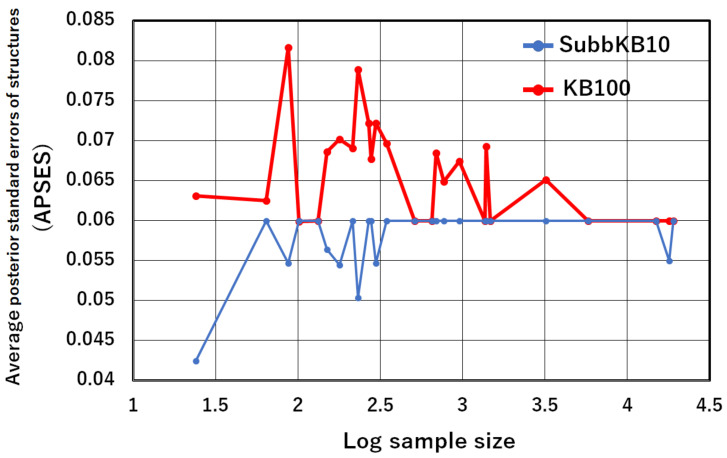
Average posterior standard errors of structures (APSES) of the *KB100* and those of *SubbKB10*.

**Table 1 entropy-24-00743-t001:** Computational environment.

CPU	Intel(R) Xeon(R) E5-2630 v4 10 Cores, 2.20 GHz
System Memory	128 GB
Software	Java 1.8

**Table 2 entropy-24-00743-t002:** Datasets for the experiments.

No.	Datasets	Sample Size	Variables	Entropy $H(X_{0})$
1	lenses	24	5	0.9192
2	mux6	64	7	0.6931
3	post	87	9	0.6480
4	zoo	101	17	1.2137
5	HayesRoth	132	5	1.0716
6	iris	150	5	1.0986
7	wine	178	14	1.0860
8	glass	214	10	1.5087
9	CVR	232	17	0.6908
10	heart	270	14	0.6870
11	BreastCancer	277	10	0.6043
12	cleve	296	14	0.6899
13	liver	345	7	0.6804
14	threeOf9	512	10	0.6907
15	crx	653	16	0.6888
16	Australian	690	15	0.6871
17	pima	768	9	0.6468
18	TicTacToe	958	10	0.6453
19	banknote	1372	5	0.6870
20	Solar Flare	1389	11	0.6073
21	CMC	1473	10	1.0668
22	led7	3200	8	2.3006
23	shuttle-small	5800	10	0.6606
24	EEG	14980	15	0.6879
25	HTRU2	17898	9	0.3062
26	MAGICGT	19020	11	0.6484

**Table 3 entropy-24-00743-t003:** Classification accuracies of each BNC.

				aCLL-				Adaboost		Bagging	Bagging		*K*B10				Subb*K*B10	
No.	$H(X_{0})$	NB	TAN	TAN	EBN	EANB	${\mathrm{bAN}}_{\mathrm{mix}}$	(EBN)	B-RAI	(EBN)	(EANB)	*K*B10	(EANB)	*K*B20	*K*B50	*K*B100	(MDL)	Subb*K*B10
1	0.9192	0.6250	0.7083	0.7083	0.8125	**0.8750**	0.6667	0.8125	0.8500	0.8333	**0.8750**	0.8333	0.6250	0.8333	0.8333	0.8333	**0.8750**	0.8333
2	0.6931	0.5469	0.6094	0.5938	0.4531	0.5469	0.5938	0.4531	0.3238	0.6094	0.4063	0.3594	0.5781	0.4219	0.4219	0.4219	0.3906	**0.6250**
3	0.6480	0.6552	0.6322	0.5977	0.7126	0.7126	0.6552	0.7126	**0.7139**	0.7126	0.7126	0.7126	0.6552	0.7126	0.7126	0.7126	0.7126	0.7126
4	1.2137	**0.9901**	0.9406	0.9505	0.9426	0.9604	**0.9901**	0.9406	0.9435	0.9604	0.9604	0.9505	0.9703	0.9505	0.9505	0.9505	0.9307	0.9505
5	1.0716	0.8106	0.6439	0.6742	0.6136	**0.8333**	0.6970	0.6136	0.6143	0.6136	**0.8333**	0.8182	0.7955	0.8182	0.8182	0.7803	0.8182	0.7727
6	1.0986	0.7133	**0.8267**	0.8200	**0.8267**	0.8067	**0.8267**	0.8200	0.8133	**0.8267**	**0.8267**	**0.8267**	**0.8267**	**0.8267**	**0.8267**	0.8200	0.8000	**0.8267**
7	1.0860	0.9270	0.9213	0.9157	0.9438	0.9270	0.9326	0.9213	0.8941	**0.9551**	0.9213	0.9438	0.9270	0.9438	0.9438	0.9438	**0.9551**	0.9438
8	1.5087	0.5421	0.5467	**0.6215**	0.5607	0.5280	0.5981	0.5701	0.5470	0.5701	0.5234	0.5701	0.5888	0.5748	0.5748	0.5748	0.5607	0.5748
9	0.6908	0.9095	0.9526	0.9224	0.9612	0.9526	0.9310	0.9655	0.9697	**0.9698**	0.9569	0.9612	0.9569	0.9655	0.9655	0.9655	0.9612	**0.9698**
10	0.6870	0.8296	0.8333	0.8148	0.8296	**0.8444**	0.8333	0.8074	0.7611	0.8407	0.8407	0.8259	0.8222	0.8333	0.8333	0.8333	0.8333	0.8370
11	0.6043	0.7365	0.7220	0.6968	0.7076	0.6751	0.7148	**0.7509**	0.6888	0.7004	0.6787	0.7040	0.7148	0.7040	0.7076	0.7329	0.7004	0.7220
12	0.6899	0.8311	0.8243	**0.8446**	0.8074	0.8142	0.8176	0.7939	0.7771	0.8108	0.8142	0.8074	0.8209	0.8041	0.8074	0.8176	0.8142	0.8176
13	0.6804	0.6464	0.6609	0.6522	0.5768	0.6058	0.6638	0.5971	0.5995	0.6174	0.6261	0.5913	**0.6783**	0.6087	0.6145	0.6261	0.6087	0.6232
14	0.6907	0.8008	0.8691	0.8906	0.8691	0.8672	0.8789	0.9063	0.7598	0.8906	0.8789	0.9043	0.8926	0.8984	0.8965	**0.9434**	0.9043	0.9023
15	0.6888	0.8392	0.8515	0.8453	0.8392	0.8622	0.8331	0.8591	0.8590	0.8499	**0.8652**	0.8392	0.8392	0.8392	0.8499	0.8484	0.8530	0.8499
16	0.6871	0.8348	0.8290	0.8478	0.8565	0.8580	0.8333	**0.8638**	0.8493	0.8464	0.8594	0.8565	0.8362	0.8565	0.8536	0.8478	0.8565	0.8464
17	0.6468	0.7057	0.7188	0.7031	0.7253	0.7188	0.7083	0.7240	0.7123	0.7227	0.7161	0.7279	0.7201	0.7279	0.7266	**0.7331**	0.7266	0.7266
18	0.6453	0.6889	0.7599	0.7192	0.8549	0.8445	0.7505	**0.9123**	0.6994	0.8466	0.8445	0.8539	0.8518	0.8518	0.8528	0.8486	0.8925	0.8518
19	0.6870	0.8433	**0.8819**	0.8761	0.8812	0.8812	0.8754	0.8776	0.8812	0.8812	0.8812	0.8812	0.8812	0.8812	0.8812	0.8812	0.8812	0.8812
20	0.6073	0.7804	0.7970	0.8200	**0.8431**	**0.8431**	**0.8143**	**0.8431**	0.8409	**0.8431**	**0.8431**	**0.8431**	0.8236	**0.8431**	**0.8431**	**0.8431**	**0.8431**	**0.8431**
21	1.0668	0.4644	0.4725	0.4650	0.4549	0.4270	**0.4779**	0.4399	0.4100	0.4521	0.4270	0.4535	0.4623	0.4542	0.4494	0.4616	0.4481	0.4487
22	2.3006	0.7288	0.7309	**0.7347**	0.7288	0.7288	0.7300	0.7288	0.7228	0.7284	0.7284	0.7288	0.7281	0.7288	0.7288	0.7303	0.7272	0.7309
23	0.6606	0.9383	0.9567	0.9538	0.9693	**0.9716**	0.9681	0.9662	0.9659	0.9693	0.9702	0.9693	0.9714	0.9693	0.9693	0.9693	0.9393	0.9693
24	0.6879	0.5774	0.6298	0.6138	0.6844	0.6895	0.6031	0.6906	0.6450	0.6881	**0.6955**	0.6857	0.6931	0.6856	0.6856	0.6885	0.6918	0.6899
25	0.3062	0.8966	**0.9141**	**0.9141**	**0.9141**	**0.9141**	0.9102	0.9073	0.9066	**0.9141**	**0.9141**	**0.9141**	**0.9141**	**0.9141**	**0.9141**	**0.9141**	**0.9141**	**0.9141**
26	0.6484	0.7447	0.7769	0.7656	0.7859	0.7879	0.7734	0.7849	0.7827	0.7859	**0.788**	0.7863	0.7877	0.7863	0.7863	0.7871	0.7855	0.7860
	Ave	0.7541	0.7696	0.7678	0.7752	0.7875	0.7722	0.7793	0.7512	0.7861	0.7841	0.7826	0.7831	0.7859	0.7864	0.7888	0.7855	**0.7942**

**Table 4 entropy-24-00743-t004:** The *p*-values of each BNC.

			aCLL-				Adaboost		Bagging			
	NB	TAN	TAN	EBN	EANB	${\mathrm{bAN}}_{\mathrm{mix}}$	(EBN)	B-RAI	(EBN)	*K*B10	*K*B20	*K*B100
*p*-values	0.0016	0.0017	0.0046	0.0013	0.0749	0.0069	0.0197	0.0001	0.0315	0.0655	0.0694	0.0617

**Table 5 entropy-24-00743-t005:** Average numbers of the class variable’s parents in the structures of the *EBN* and those of the *SubbKB10*.

No.	Datasets	Sample Size	Variables	*EBN*	Subb*K*B10
1	lenses	24	5	0.90	1.37
2	mux6	64	7	5.70	4.68
3	post	87	9	0.00	0.02
4	zoo	101	17	3.70	4.31
5	HayesRoth	132	5	3.00	2.46
6	iris	150	5	1.80	1.89
7	wine	178	14	1.70	1.40
8	glass	214	10	0.40	0.68
9	CVR	232	17	0.90	1.42
10	heart	270	14	1.70	1.54
11	BreastCancer	277	10	0.70	0.82
12	cleve	296	14	1.90	1.69
13	liver	345	7	0.00	0.19
14	threeOf9	512	10	5.00	3.85
15	crx	653	16	1.20	1.08
16	Australian	690	15	1.00	1.14
17	pima	768	9	1.60	1.09
18	TicTacToe	958	10	1.60	0.40
19	banknote	1372	5	0.00	0.69
20	Solar Flare	1389	11	0.80	0.91
21	CMC	1473	10	0.90	0.82
22	led7	3200	8	0.60	0.95
23	shuttle-small	5800	10	2.00	2.12
24	EEG	14980	15	0.50	0.47
25	HTRU2	17898	9	1.50	1.62
26	MAGICGT	19020	11	0.00	0.47
	Average			1.50	1.46

**Table 6 entropy-24-00743-t006:** Classification accuracies of *K*-best EC methods and *SubbKB10* for 1/100-sized subsamples from MAGICGT.

	EBN	*K*B10	*K*B20	*K*B50	*K*B100	Subb*K*B10
Accuracy	0.7498	0.7509	0.7529	0.7557	0.7563	**0.7579**

**Table 7 entropy-24-00743-t007:** Average SHDs of model averaging BNCs.

	Bagging	Bagging		*K*B10		
No.	(EBN)	(EANB)	*K*B10	(EANB)	*K*B100	Subb*K*B10
1	1.07	0.33	2.61	2.30	3.09	4.11
2	0.61	0.89	3.24	1.96	4.33	4.56
3	0.42	0.42	2.33	1.92	2.31	3.32
4	32.56	12.98	7.03	7.41	5.43	10.51
5	0.00	0.07	2.35	2.39	2.87	3.78
6	1.87	1.54	5.09	2.86	4.23	6.12
7	11.85	5.50	7.57	2.56	4.04	9.40
8	4.76	5.31	3.71	4.33	3.33	5.36
9	23.27	24.25	6.37	6.60	3.03	8.91
10	7.19	6.42	4.48	2.09	3.61	7.64
11	1.02	1.02	2.36	2.20	0.76	4.67
12	5.95	5.07	3.74	2.16	2.50	7.34
13	4.27	4.17	4.98	2.08	4.45	6.63
14	4.33	3.36	3.16	3.44	2.59	3.79
15	8.68	6.29	9.36	3.70	7.18	10.63
16	10.79	9.39	6.25	3.97	5.60	9.82
17	3.30	1.97	5.05	3.15	4.63	7.10
18	8.06	5.85	7.86	7.18	7.19	10.54
19	0.00	0.00	5.54	3.74	3.77	6.88
20	3.24	2.58	5.20	4.38	4.10	7.19
21	4.64	3.60	5.57	2.79	3.63	6.67
22	0.00	0.00	3.58	1.80	1.21	5.49
23	1.80	5.05	6.23	5.83	5.56	6.87
24	7.99	15.40	9.04	12.07	6.92	10.26
25	0.32	0.32	6.03	5.01	0.80	9.56
26	3.82	1.36	9.02	7.14	4.28	12.47
Ave	4.98	4.16	4.82	3.64	3.72	6.70

**Table 8 entropy-24-00743-t008:** (1) Average posterior standard errors of structures (APSES) of the *KB100* and those of the *SubbKB10* and (2) classification accuracies of *KB100* and the *SubbKB10*.

				(1) APSES		(2) Classification Accuracy	
No.	Datasets	Sample Size	Variables	*KB100*	*SubbKB10*	*KB100*	*SubbKB10*
1	lenses	24	5	0.0631	0.0425	0.8333	0.8333
2	mux6	64	7	0.0625	0.0600	0.4219	0.6250
3	post	87	9	0.0817	0.0547	0.7126	0.7126
4	zoo	101	17	0.0599	0.0600	0.9505	0.9505
5	HayesRoth	132	5	0.0600	0.0600	0.7803	0.7727
6	iris	150	5	0.0686	0.0564	0.8200	0.8267
7	wine	178	14	0.0702	0.0545	0.9438	0.9438
8	glass	214	10	0.0691	0.0600	0.5748	0.5748
9	CVR	232	17	0.0789	0.0504	0.9655	0.9698
10	heart	270	14	0.0722	0.0600	0.8333	0.8370
11	BreastCancer	277	10	0.0677	0.0600	0.7329	0.7220
12	cleve	296	14	0.0722	0.0547	0.8176	0.8176
13	liver	345	7	0.0697	0.0600	0.6261	0.6232
14	threeOf9	512	10	0.0600	0.0600	0.9434	0.9023
15	crx	653	16	0.0600	0.0600	0.8484	0.8499
16	Australian	690	15	0.0685	0.0600	0.8478	0.8464
17	pima	768	9	0.0649	0.0600	0.7331	0.7266
18	TicTacToe	958	10	0.0674	0.0600	0.8486	0.8518
19	banknote	1372	5	0.0600	0.0600	0.8812	0.8812
20	Solar Flare	1389	11	0.0693	0.0600	0.8431	0.8431
21	CMC	1473	10	0.0600	0.0600	0.4616	0.4487
22	led7	3200	8	0.0651	0.0600	0.7303	0.7309
23	shuttle-small	5800	10	0.0600	0.0600	0.9693	0.9693
24	EEG	14980	15	0.0600	0.0600	0.6885	0.6899
25	HTRU2	17898	9	0.0600	0.0550	0.9141	0.9141
26	MAGICGT	19020	11	0.0600	0.0600	0.7871	0.7860
	Average			0.0658	0.0580	0.7888	0.7942
	*p*-value			0.0001	-	-	-

**Table 9 entropy-24-00743-t009:** Classification accuracies of XGBoost, CatBoost, LightGBM, and *SubbKB10*.

No.	Datasets	Sample Size	Variables	$H(X_{0})$	XGBoost	CatBoost	LightGBM	*Subb* *K* *B10*
1	lenses	24	5	0.9192	0.7833	0.7833	0.6667	0.8333
2	mux6	64	7	0.6931	0.8333	0.9857	0.5357	0.8281
3	post	87	9	0.6480	0.6806	0.6000	0.7139	0.7011
4	zoo	101	17	1.2137	0.9509	0.9409	0.9309	0.9505
5	HayesRoth	132	5	1.0716	0.7967	0.7956	0.7429	0.8258
6	iris	150	5	1.0986	0.8200	0.8267	0.8200	0.8267
7	wine	178	14	1.0860	0.9268	0.9373	0.9088	0.9157
8	glass	214	10	1.5087	0.6457	0.6604	0.6407	0.6402
9	CVR	232	17	0.6908	0.9656	0.9612	0.9612	0.9612
10	heart	270	14	0.6870	0.8370	0.8111	0.8185	0.8259
11	BreastCancer	277	10	0.6043	0.7390	0.7361	0.7394	0.6931
12	cleve	296	14	0.6899	0.8277	0.7940	0.8172	0.8311
13	liver	345	7	0.6804	0.6635	0.6434	0.6548	0.6174
14	threeOf9	512	10	0.6907	1.0000	1.0000	1.0000	0.9980
15	crx	653	16	0.6888	0.8589	0.8697	0.8513	0.8637
16	Australian	690	15	0.6871	0.8623	0.8565	0.8609	0.8507
17	pima	768	9	0.6468	0.7136	0.7188	0.7149	0.7018
18	TicTacToe	958	10	0.6453	1.0000	1.0000	1.0000	0.9979
19	banknote	1372	5	0.6870	0.8812	0.8812	0.8812	0.8812
20	Solar Flare	1389	11	0.6073	0.8402	0.8359	0.8186	0.8409
21	CMC	1473	10	1.0668	0.4894	0.4684	0.4725	0.4807
22	led7	3200	8	2.3006	0.7297	0.7309	0.7303	0.7281
23	shuttle-small	5800	10	0.6606	0.9721	0.9721	0.9721	0.9722
24	EEG	14980	15	0.6879	0.7376	0.7308	0.7348	0.8901
25	HTRU2	17898	9	0.3062	0.9141	0.9141	0.9141	0.9141
26	MAGICGT	19020	11	0.6484	0.7871	0.7863	0.7870	0.7855
	Average				0.8176	0.8169	0.7957	0.8213
	*p*-value				p>0.1	p>0.1	p>0.1	-

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
