# Peer review of "Bayesian Network Model Averaging Classifiers by Subbagging"

_entropy, 2022, doi:10.3390/e24050743_

Round 1

Reviewer 1 Report

1. 
The text needs minor corrections which are annotated in the attached file. 

2. 
The paper is about Bayesian Networks, but there are no examples of Bayesian Networks in the methodology and in the experiments. 

3.
It is not clear to me what is the effective contribution of the paper.
The proposed method is limited to section 4 (one page). 
Subbagging is the core of the methodology, but subbagging is already proposed in [26].
Moreover in the acknowledgements the authors say:
"Parts of this research were reported in an earlier conference paper published by Sugahara et al. [54]".
The authors should extend or highlight their original contribution in the paper. 
The authors should clearly indicate which parts of the work are already published in [54] and other works, and which parts are completely new. 

4.
In the experiments the authors apply several methods. 
The authors should highlight the results obtained by the proposed method. 
For example, in the tables, the results obtained by applying their method should be evidenced by using a bold font or a different colour. 

5.
Since subbagging is mentioned in the title of the paper, in the abstract the authors should briefly explain what subbagging is. 
In this way, the abstract will give a more complete idea of the contribution of the paper.

6. 
I suggest writing some lines at the end of the introduction, about the contents of the following sections. 

7. 
The paper contains repetitions. 
Concepts like BNC, CLL, ANB, etc., are described both in section 1 and in section 2 with similar sentences.
For example, in section 1, the authors say:

"they proposed exact learning augmented naive Bayes (ANB) classifier, in which the class variable has no parent and in which all feature variables have the class variable as a parent. 

Then in section 2, the authors say:

"Friedman et al. [8] proposed the augmented naive Bayes (ANB) classifier, for which the class variable has no parent and for which all feature variables have the class variable as a parent". 

Another example of repetition is in section 1 where they say:

"the class variable tends to have numerous parents when the sample size is small. Therefore, the conditional probability parameter estimation of the class variable becomes unstable because the number of parent configurations becomes large. Then the sample size for learning a parameter becomes sparse. This analysis suggests that exact learning BNC by maximizing the ML to have no parents of the class variable might improve the classification accuracy. Consequently, they proposed exact learning augmented naive Bayes (ANB) classifier, in which the class variable has no parent and in which all feature variables have the class variable as a parent. Additionally, they demonstrated the effectiveness of their method empirically".

Then in section 2 they say:

"The class variable tends to have numerous parents for a small sample. Therefore, estimation of the conditional probability parameters of the class variable becomes unstable because the number of parent configurations be-
comes large. Then the sample size for learning a parameter becomes sparse. This analysis suggests that exact learning BNC by maximization of the ML to have no parent of the class variable might improve the classification accuracy. Consequently, they proposed exact learning ANB because the class variable has no parent in ANB structures. Additionally, they demonstrated the effectiveness of their method empirically".

And so on. 
I suggest removing all the repetitions. 
A concept can be simply mentioned or briefly explained in section 1; then the details can be described in section 2. 

8. 
The paper contains one figure. 
Is it possible to add more figures?
For instance, examples of Bayesian networks, or other diagrams summarizing the results?

9.
The paper contains many acronyms. 
In order to help the reader, the list of the acronyms and their meaning, would be helpful at the begin of the paper.

10.
In case of resubmission of the paper, I suggest evidencing the changes to the text by means of a different colour.
This will help the reviewers' work.

Reviewer 2 Report

Questions asked by Shouta Sugahara, Itsuki Aomi and Maomi Ueno are particularly interesting. The case of not too large dataset is quite complex.

The introduction is well written, very explanatory. It would be nice to have some examples of real application of BNC. Also, the K-best is classic, but isn't the K also a bit sensitive? 

Is alpha=1.0 really applicable all the time (and especially to small, size data sets).

"by experimentation that the BNC performance achieved by maximizing the ML is not necessarily worse than that achieved by maximizing CLL for small data. "But doesn't this depend on the nature of the dataset?

"The Bayesian Dirichlet equivalence uniform (BDeu), as proposed by Buntine, is often used." and it is the case here, but shouldn't we test other ones to know if it is not the most appropriate here?

The different models are clearly exposed and explained, the authors are specialists in these questions. However, the presentation of the experiments is more difficult to follow.

It is nice to have 26 different datasets, but it is also linked with no explanation and presentation of them. Some seems to have been bins-erzed. In few words, some must be presented and also teir results discussed. Do they all have ame type of distributions?

Similarly, it would have been interesting to have big datasets sampled as small datasets, i.e. 19020 of MAGICCT could have been used as 1/10th or 1/20th  and tested to see the robustness of the results.

Some Tables could have been translated into Figures to be more explicative, i.e. Table 3 (or at least with less digits).

Table 3 is a good example of the difficulty to see at the different datasets, number 2 is really difficult while number 4 is highly simple.

The term ‘proposed method’ or ‘proposal’ is not really nice. You can perhaps add a new name in the text.

Table 5. Is n°13 at 0.00? How is it possible?

Conclusion is very well presented and perspective very interesting.

Reviewer 3 Report

The paper proposes a subbaging strategy to improve the classification accuracy of Bayesian network classifier. Overall, the paper's structure is good and the design of the experiment has followed the standard machine learning benchmark, where the proposed model is validated on multiple datasets against the multiple competitors. However, the paper contains some weaknesses that should be improved, particularly the clarity of the paper should be revised.

  • The validation method seems one-time holdout. It would be better to run multiple hold-out (e.g., subsampling) or cross-validation so that the result is not obtained by chance.
  • Table 3 is not informative and unclear. Use bold to indicate the best result. Using bigger font would be also helpful for the readers.
  • The statistical significance tests should be done via two-step tests, i.e., omnibus test and posthoc test. For instance, the authors might consider the Friendman test then followed by the Nemenyi test. Hommel test is not clearly explained how it works?
  • Figure 1 is not clear and hard to understand.
  • I do not particularly agree to compare an ensemble model against deep neural network. Why don't you compare it with state-of-the-art ensemble models such as XGBoost, CatBoost, and LightGBM? Deep neural network has not shown a significant performance on tabular data! It received a lot of criticism, particularly when dealing with tabular data.

Round 2

Reviewer 1 Report

My suggestions have been applied.

The paper can be accepted as it is.

Reviewer 2 Report

The authors answered all of my questions with great efficiency. It is a pleasure to read both the manuscript and the answer letter. The manuscript has been greatly improved.

Reviewer 3 Report

I think the authors have addressed all previous comments accordingly, except for the omnibus test which is not done yet. Overall, the paper is acceptable as it stands now.